# Fetoscopic Myelomeningocele Repair with Complete Release of the Tethered Spinal Cord Using a Three-Port Technique: Twelve-Month Follow-Up—A Case Report

**DOI:** 10.3390/diagnostics12122978

**Published:** 2022-11-28

**Authors:** Agnieszka Pastuszka, Mateusz Zamłyński, Tomasz Horzelski, Jacek Zamłyński, Ewa Horzelska, Iwona Maruniak-Chudek, Adrianna Marzec, Justyna Paprocka, Patrycja Gazy, Tomasz Koszutski, Anita Olejek

**Affiliations:** 1Department of Gynecology, Obstetrics and Oncological Gynecology, Bytom, Medical University of Silesia, 40-752 Katowice, Poland; 2Department of Pediatric Surgery and Urology, Faculty of Medical Sciences in Katowice, Medical University of Silesia, 40-752 Katowice, Poland; 3Department of Neonatology and Neonatal Intensive Care, Medical University of Silesia, 40-752 Katowice, Poland; 4Department of Pediatric Neurology, Medical University of Silesia, 40-752 Katowice, Poland

**Keywords:** spina bifida, myelomeningocele, prenatal surgery, fetoscopy

## Abstract

Open spina bifida is one of the most common congenital defects of the central nervous system. Open fetal surgery, which is one of the available therapeutic options, remains the gold standard for prenatal repairs. Fetoscopic closure may lower the number of maternal complications associated with open fetal surgery. Regardless of the approach, the outcome may be compromised by the development of tethered spinal cord (TSC) syndrome. At 24.2 weeks of gestation, a primipara was admitted due to fetal myelomeningocele and was deemed eligible for fetoscopic repair. Fetal surgery was performed at 25.0 weeks of gestation. It was the first complete untethering of the spinal cord and anatomic reconstruction (dura mater, spinal erectors, skin) achieved during a fetoscopic repair of spina bifida. Cesarean section due to placental abruption was performed at 31.1 weeks of gestation. VP shunting, with no need for revision, was performed at 5 weeks postdelivery due to progressing ventriculomegaly. No clinical or radiological signs of secondary tethering were observed. Neurological examination at 11 months postdelivery revealed cranial nerves without any signs of damage, axial hypotonia, decreased muscle tone in the lower extremities, and absent pathological reflexes. Motor development was slightly retarded. Complete untethering of the neural structures should always be performed, regardless of the surgical approach, as it is the only course of action that lowers the risk for developing secondary TSC.

## 1. Introduction

Open spina bifida (OSB) is a non-lethal spinal dysraphism, developing between day 21 and 28 of the embryonic life [1]. Knowledge of the non-genetic and population risk factors (low intake of folic acid; use of anti-epileptic drugs; diabetes and obesity) for the development of neural tube defects allows implementation of preventive measures before conception [2]. The incidence of spina bifida has been estimated at 3–4/10,000 live births [3,4].

Until 1997, surgical correction of the defect within the first 48 h of neonatal life remained the only method for OSB repair. However, the treatment was associated with progressing intrauterine development of Chiari type II malformation, which in turn led to progressing fetal ventriculomegaly and hindbrain herniation [5]. Persistent exposure of the spinal nerves and the placode to the toxicity of the amniotic fluid and mechanical injury associated with intrauterine fetal movements causes the loss of motor function in the lower extremities and the development of a neurogenic bladder and bowel syndrome [5]. In addition, postnatal OSB repair is associated with the development of shunt-dependent hydrocephalus in 80% of the cases, requiring either ventriculoperitoneal (VP) shunting, to drain the excess cerebrospinal fluid (CSF) from the brain ventricles and transport it to the peritoneal cavity, or an endoscopic third ventriculostomy (ETV) [5]. These methods are associated with the risk for additional complications, such as central nervous system infections or rapidly progressing secondary hydrocephalus connected with the malfunction of the newly created route of CSF evacuation. Other, severe complications such as intraventricular hemorrhage, proximal occlusion of the shunting system (intraventricular or peritoneal), or formation of a pseudocyst in the peritoneal cavity caused by inadequate peritoneal CSF absorption capacity may appear later in a child’s life [6].

Towards the end of the previous century, Tulipan and Bruner, of Vanderbilt University in Nashville, published their reports on fetoscopic repair of open spina bifida. Bruner et al. performed an endoscopic apposition of a maternal skin graft over the spina bifida site, covering the exposed nerve structures of the placode and spinal nerves [7]. However, frustrating outcomes—only two children survived and the remaining required ventriculoperitoneal shunting—were the reason the method was temporarily abandoned [8]. In 2011, a randomized controlled clinical trial (MOMS study) compared the outcomes of prenatal open surgery with postnatal OSB repair (91 and 91 patients, respectively) [9]. The results demonstrated that the need for ventriculoperitoneal shunting was significantly lower in the prenatal as compared to the postnatal repair groups (40% and 82%, respectively; *p* = 0.001). Moreover, the number of infants who were able to walk without orthoses/braces or other assistive orthopedic devices was two-fold higher in the prenatal as compared to the postnatal repair groups (42% vs. 21%, respectively). The composite score for mental development and motor function at 30 months postdelivery was also markedly higher in the prenatal repair group. During the first year of life, 36% of the infants from the prenatal group presented no symptoms of hindbrain herniation as compared to 84% in the postnatal repair group. The improvement in the neurological development in the randomized MOMS trial was limited by the occurrence of the TSC (tethered spinal cord) syndrome [9]. TSC remains a risk factor for less favorable neurological outcomes in the pre- and postnatal repair groups, regardless of the surgical approach (fetoscopic vs. open surgery) [10,11]. Our original technique of complete untethering of the spinal cord and the use of fetal native tissue (dura mater, spinal erectors, skin) for anatomic reconstruction reduce the risk of secondary TSC.

## 2. Case Study

A primipara (Caucasian, BMI 28.3 kg/m^2^, no history of drinking and/or drug abuse, no co-existing morbidities.) at 24.2 weeks gestation was admitted to our clinic in May of 2021. Fetal ultrasound revealed the lemon sign, normal division of the forebrain, normal cerebral falx; width of the lateral ventricles: AD left side 19.6, AD right side 15.7; CSP—4.8 mm, cisterna magna—3.5 mm, no signs of ACC. Hindbrain ultrasound revealed the banana sign and stage I herniation (Sutton Scale) [12]. Spinal imaging demonstrated a 14 mm long myelomeningocele (MMC), upper point of the defect was positioned at L_4_, with a hernia sack (22 × 19 mm). Axial view found the lower extremity joints (hip, knee and tarsal) to be normal, with no signs of genu varum or genu valgum; motor function of all three joints was preserved. Amniotic fluid volume was normal (AFI 10 cm) and the placenta was located on the anterior wall. The estimated fetal weight (EFW) according to Hadlock’s formula (BPD-HC-AC-FL) was 470 g/50 pct. (Figure 1).

Fetal karyotype was 46 XY. Whole-genome oligonucleotide microarray (CytoSure Constitutional v3 (8 × 60 k), Oxford Gene Technology, GRCh37/hgl9/ Oxfordshire, United Kingdom)) arr(l-22) × 2,(X,Y)xl was normal. Physical examination revealed risk factors for venous thromboembolism: varicose veins of the lower limb with normal deep vein flow on Doppler ultrasound, presence of homozygous PAI-1 gene, family (first-degree) history of stroke. Caprini score was 8 points.

## 3. Preoperative Procedures

MRI SSFSET2 (T2-Weighted, Single Shot, Fast Spin Echo Magnetic Resonance Imaging) was performed and revealed the presence of a myelomeningocele (MMC), situated between L_2_ to S_2_. The length and the width of the split were 9.8 and 4.2 mm, respectively. Hernia sack measurements were: 27 mm × 22 mm × 19 mm. Severe ventriculomegaly AD 18.1 mm was found. Posterior cranial fossa with signs of herniation (Chiari II) to the spinal cervical canal (8.6 mm), stage II Sutton Scale [12]. No ACC was found (Figure 2).

The inclusion and exclusion criteria from the MOMS protocol were used [9]. An interdisciplinary team explained the nature of the fetal health problem to the mother. The available surgical repair methods, both pre- and postnatal, were presented. Complications were discussed at length, including the necessity of conversion to open fetal surgery. Informed maternal consent for a prenatal OSB repair was obtained. The local Ethics Committee approved the procedure (no. L.dz.NN-013296/I/02/03).

Preoperatively, the patient received nifedipine—3 × 10 mg/24 h/p.o, progesterone—2 × 200 mg/24 h/p.vag. indomethacin—150 mg/24 h/p.o. and cefazoline—2 × 1 g/24 h/i.v. Enoxaparin sodium—40 mg/24 h/s.c and compression therapy (stage II for 48 h) were used as antithrombotic measures.

## 4. Fetoscopic Myelomeningocele Repair

Fetoscopic MMC repair was performed at 25.0 weeks gestation. Sevoflurane (2:1) was used for maternal inhalational anesthesia (no other tocolytic agents were used), with additional continuous epidural anesthesia. The tocolytic lines and anesthesia procedures were the same as those during open fetal surgery, as described in the protocol [13]. Laparotomy with lower-segment transverse incision was performed and the uterus was exteriorized. Ultrasound was used to determine the exact position of the placenta on the anterior wall. The 3.5 mm ports were inserted into the uterus using the Belfort technique, with our own modifications [14]. The optical port was inserted through all layers of the uterine wall, including the amniotic membrane, and secured with two opposing sutures The same technique was used for the two surgical ports. A diode laser beam was used to separate the muscle (2 mm length), up to the level of the amniotic membrane. Humidified CO_2_ heated to 37.7 °C was infused into the amniotic cavity while the amniotic fluid was gradually removed. Adequate pressure inside the amniotic cavity was maintained using carbon dioxide insufflation (0.6 L/min; range: 12–15 mm Hg) to prevent uterine contractility.

## 5. Open Fetoscopic Myelomeningocele Repair

An opioid (Fentanyl 20 mcg/kg) with a muscle relaxant (Rocuronium 0.3 mg/kg) was injected (intramuscular) into the fetal buttocks using a needle passed through one of the ports. Traction suture (Vicryl 3.0) was placed through the uterine wall to stabilize fetal position. MMC closure was performed in layers and normal anatomy was restored. The hernia sack was opened and all visible neural structures were separated from the sack wall. Dura mater, muscle, and skin layers were symmetrically separated. The spinal cord was separated from the arachnoid membrane between the placode and the junctional zone [Belfort]. The dural-periosteal ligament (adhesion) was incised at the upper part (i.e., the porta) of MMC, which allowed the release of the tethered placode. Dural tissue was separated and placed over the exposed placode, achieving watertight coverage. Next, spinal erectors were separated and symmetrically released on both sides, along the entire length of the MMC. Complete release of the paravertebral muscles allowed repositioning them at the posterior median line and to cover the entire MMC length. The muscles were repositioned to their anatomical position and sutured along the posterior median line using a continuous suture (Stratafix 4.0 Ethicon US). At the last stage of the closure, the skin covering the muscles, with the subcutaneous tissue, was separated and released. The edges of the skin and the subcutaneous tissue, tension-free, were moved to the posterior median line and sutured using a continuous suture. The traction suture was removed after the repair. Perioperatively, fetal wellbeing was monitored using Doppler ultrasound; no PI < 1 and FHR < 115/min were observed in UA (Figure 3). The amniotic cavity was infused with heated 0.9% NaCl crystalloid solution to achieve preoperative AFI values and 1.0 g of cefazoline was infused to the amniotic fluid. The ports were removed and single sutures were placed through all layers of the uterine wall to prevent chorioamniotic membrane separation (CAS). Operative time (skin to skin) was 220 min.

## 6. Postoperative Monitoring

For the first two postoperative days, the inflammatory markers (CRP, maternal blood leukocytosis, complete blood count, coagulation due to risk for thrombosis) were closely monitored. The mother received indomethacin 150 mg p.o./24 (only 1 day) and nifedipine 40 mg/24 h; antibiotic prophylaxis was continued with Cefazoline 2 × 1 g iv until day 5; and enoxaparin 0.4 mg/24 h was received until delivery. No tocolytics were used. Fetal wellbeing was monitored every day by measuring the UA Doppler flow. Serial amniotic fluid index was AFI: 7–9 cm. Uterine tone on CTG was 10–12 mm Hg. At 30 weeks of gestation, fetal parameters were as follows: EFW 1058 g/50 pct.; on neuroscan: BPD 82 mm, HC 293 mm, severe ventriculomegaly AD 18.1 mm; reversed hindbrain herniation (00 Sutton Scale); lower extremity motor function was preserved.

## 7. Child Development up to 12 Months of Age

Cesarean section was performed at 31.1 weeks of gestation due to placental abruption. Uterine fetoscopic scars after the fetoscopic access were fully healed. A boy (1550 g 8/8Apgar score) was delivered. The MMC scar did not require correction. Respiratory support with nCPAP and caffeine therapy were needed due to sporadic episodes of apnea of prematurity with concomitant moderate bradycardia with desaturation, which are the consequences of prematurity. The child was discharged home on day 14 postdelivery. After discharge, a follow-up fontanelle ultrasound was performed once a week. At week 5, due to progressing ventriculomegaly (AD > 20 mm) and increasing head circumference (>90 pct), a ventriculoperitoneal (VP) shunt was placed, with no need for revision. Shunting successfully stopped the development of hydrocephalus and achieved moderate ventriculomegaly (AD: 15 mm) until 12 months postdelivery. Voiding cystourethrography was performed on day 10 postdelivery and revealed no signs of the vesicoureteral reflux. On day 14, routine management was implemented: clean intermittent catheterization (CIC) every 3 h, with a 6 h break at night. Anticholinergic therapy included Ditropan (1 × 1.25 mg/night), antibiotic prevention until the end of month 4 postdelivery: Hiconcil (1 × 25 mg/kg/night), followed by Furaginum (1 × 12.5 mg/night).

Neurological examination at 11 months postdelivery revealed: head circumference of 43.5 cm, closing frontal fontanelle, cranial nerves without any signs of damage, axial hypotonia, decreased muscle tone in the lower extremities, tendon reflexes present apart from ankle reflexes, absent pathological reflexes. Psychomotor development of the child is slightly retarded.

At 12 months postdelivery, the child can roll, sit up and continue sitting unassisted, crawl, and stand up while holding onto furniture. Psychological evaluation was normal. No history of urinary tract infections and constipation. A micturition diary revealed signs of urinary retention (10–40 mL of residual urine). To date, the child has made no attempts at controlled micturition. A urodynamic test revealed a neurogenic bladder of normal volume, and unstable detrusor with hyper-reflexia (Figure 4).

High-pressure bladder and bladder capacity correlated with age (80 mL). Leak Point Pressure at 25 mL volume: Pdet/Pves to 48 cm H_2_O, at 38 mL volume: Pdet/Pves to 60 cm H_2_O.

After urodynamic testing at 12 months postdelivery, CIC four times a day and continuation of the anticholinergic therapy (Ditropan 2 × 1.25 mg) were recommended. The child does not require preventive antibiotic therapy.

Ultrasound test of the bifid spine revealed a mobile medullary cone, positioned at L_3_. No clinical signs of tethering were observed. The VP shunt was patent at 11 months postdelivery (Evans index 0.35) (Figure 5).

## 8. Discussion

Open spinal dysraphism is a complex defect that develops during the initial stages of the embryonic life and results in the development of Chiari type II malformation [1]. Mid- and hindbrain abnormalities remain the key signs of damage to the central nervous system of the developing fetus [10]. Prenatal imaging, both MRI and ultrasound, in fetuses with OSB typically reveal symptoms of colpocephaly, compressed structures of the posterior fossa with tonsil, cerebellum, and afterbrain herniation (hindbrain herniation—HBH) into the foramen magnum [15]. According to the hypothesis of McLone and Knepper, these abnormalities are associated with severe CSF leak through the OSB site [16].

Recurrent TSC after primary pre- or postnatal OSB repair is a polysymptomatic complication which is characterized by neurologic, orthopedic, and urologic manifestations [17]. It is caused by insufficient untethering of the neural structures during the repair and secondary tethering of the placode and the spinal nerves in the postoperative scars [17]. Secondary fixed low position of the medullary cone leads to mechanical damage to the overstretched neural structures during fetal and neonatal life [17]. In consequence, due to being unable to migrate upwards to the physiological upper portion of the spinal cord and spinal nerves, inflammatory, metabolic, and vascular abnormalities develop within these structures. The incidence of TCS has been estimated at 10–30% [18,19].

A randomized MOMS trial was the first study to demonstrate a comparable reoperation rate in children aged up to 12 months due to secondary TCS in the prenatal (6/77 (8%) and postnatal (1/80 (1%) OSB repair groups (RR6.15; 0.76–50.00; p.06) [9]. Verbeek et al. reported that 5/13 (38%) children in the prenatal repair group required a neurosurgical intervention due to TSC as compared to 2/13 (15%); P.37 in the postnatal group; P.37 [20].

Worley et al. compared the results of fetal (*n* = 248) and postnatal (*n* = 698) interventions in cohorts of children up to 12 months of age and found that fetal surgery was associated with lower incidence of CSF diversion for hydrocephalus by VP shunting or endoscopic third ventriculostomy as compared to postnatal surgery (110/239 (46) vs. 349/441 (79) respectively; IRR 0.61; CI95%; 0.53–0.71; *p* < 0.01). Repeat tethered cord release was necessary in 54/298 (18%) vs. 102/648 (16%) cases, respectively (IRR 1.22 (0.94–1.59); CI95% 0.13; 1.11 0.84–1.47; p. 46) [11]. Low position of the medullary cone remains the main diagnostic marker for a tethered spinal cord on prenatal imaging [21]. The eligibility criteria for prenatal (fetoscopic or open surgery) OSB repair include longitudinal and transverse measurements of the split, and the presence and size of the hernia sack. An optimal OSB repair should restore the natural layers and anatomy, achieving complete correction, as during a ‘single-step’ surgery [22]. Spherical shape of the split, together with a small, flat hernia sack or no hernia sack, creates a favorable environment for TCS. Such conditions usually impede complete untethering of the neural structures and a full three-layer reconstruction of normal anatomy [23]. Technical challenges associated with the procedure result from changes in the dural structure and chronic inflammation of the subarachnoid space due to severe tissue adhesion. Pastuszka et al. evaluated 87 (*n* = 45 prenatal vs. *n* = 42 postnatal) samples of dura matter and skin harvested during the primary MMC repair. The sample analysis revealed that lymphocytic and granulocytic infiltration in the skin and the dura mater were statistically significantly more prevalent in children who underwent postnatal as compared to prenatal repair for myelomeningocele [24].

The first casuistic report of Belfort et al., about their fetoscopic myelomeningocele repair, demonstrated that a single layer of sutures to close the dura mater, muscles, and skin was associated with a CSF leak after birth, necessitating the use of Steri-Strips to treat scar dehiscence [25]. In our study, primary complete release of the tethered spinal cord consisted of symmetrical, wide separation of the dura mater, with separation of the subarachnoid space and of the muscle tissue from the dura mater, which gave us free dural patches to cover the exposed placode. The next vital step of the surgery was to free the placode and the dura mater in the porta site. Those two steps helped to fully untether the spinal cord at the porta site, allowing for physiological movement of the neural structures in the spinal canal during fetal and postnatal development. Careful and symmetrical separation of the spinal erectors, located to the side of the bifid spine, without damaging their structure, made it possible to reposition them and restore normal anatomy. That, in turn, not only allowed the protection of the spinal structures that had been outside the skeletal part of the spinal canal, but also lowered the risk for scoliosis in the future. Covering the entire length of the defect with an uncompromised, tension-free muscle decreases the risk for the development of secondary myelomeningocele shortly after surgery. It is important to bear in mind that all spinal muscles form symmetrical pairs. Non-symmetrical, inaccurate, or incomplete separation of the spinal muscles during OSB repair, pre- or postnatal, will significantly increase the risk for scoliosis in the affected child. Gentle skin separation, without the need for excessive stretching at the last stage of the surgery, lowers the risk for damage and tearing during suturing. Relaxing incisions, described by Belfort et al., should be considered in the absence of tissue relaxation [14]. Apposition of the skin edges with too much tension carries the risk for secondary wound dehiscence, which may require revision repair postdelivery [26]. Since the instruments used during a fetoscopic repair are fragile and small, the method is not without limitations. It may be successfully applied in fetuses with MMC, with a well-demarcated hernia sack and normal paravertebral muscles on diagnostic imaging. Such anatomic conditions allow for a safe and accurate separation of all tissues in all layers, resulting in fully restored anatomy.

In case of a defect with an absent or small hernia sack and concomitant weakly developed paravertebral muscles, open fetal MMC repair is recommended to allow for better separation of the tissues, without the risk of tissue injury.

## 9. Conclusions

Prenatal OSB repair using the fetoscopic approach to insert the ports into the exteriorized maternal uterus may be an effective method of treating myelomeningocele. Application of two opposing sutures through all layers of the uterine wall, including the amniotic membrane, to the site of port insertion allows for a safe and secure closure of the uterine wall once they are removed, and decreases the risk for postoperative fetal membrane separation. Regardless of the prenatal closure technique (fetoscopy, open fetal surgery), complete untethering of the neural structures (placode, spinal nerves) should always be performed, as it is the only course of action that lowers the risk of secondary tethering of the spinal cord.

Presentation of only one case is an obvious limitation of the conclusions. However, the satisfactory outcome of the introduced fetoscopic technique of complete untethering of the spinal cord and anatomic reconstruction using native fetal tissue (dura mater, spinal erectors, skin) provides good grounds for future application of this procedure.

## Figures and Tables

**Figure 1 diagnostics-12-02978-f001:**
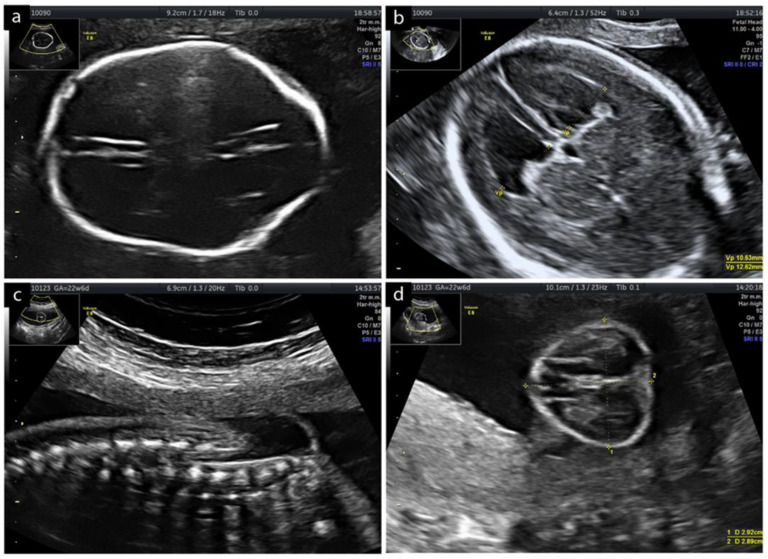
Chiari II malformation visualized on ultrasound at 22.6 weeks of gestation: (**a**) Transverse view of the fetal head: ‘lemon sign’; (**b**) Axial transventricular view of the fetal brain: the calipers indicate the measurements of the ventricular width; (**c**) TSC medullary cone positioned at L_5_; (**d**) Transverse view: MMC measurements and placode.

**Figure 2 diagnostics-12-02978-f002:**
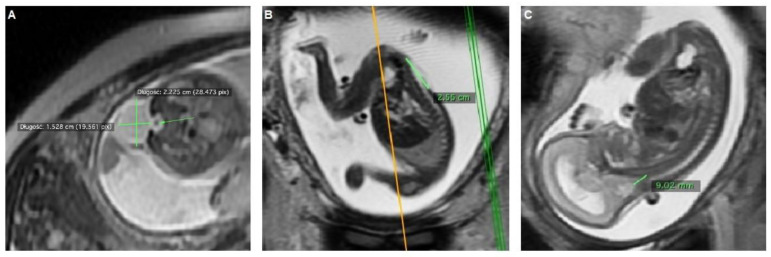
Assessment of Chiari II malformation on fetal MRI at 24.2 weeks of gestation: (**A**) Transverse view: end of the medullary cone at L_5_, MMC measurements; (**B**) Longitudinal view, MMC length, upper level of the defect: L_2_; (**C**) Sagittal view: HBH stage II (Sutton Scale).

**Figure 3 diagnostics-12-02978-f003:**
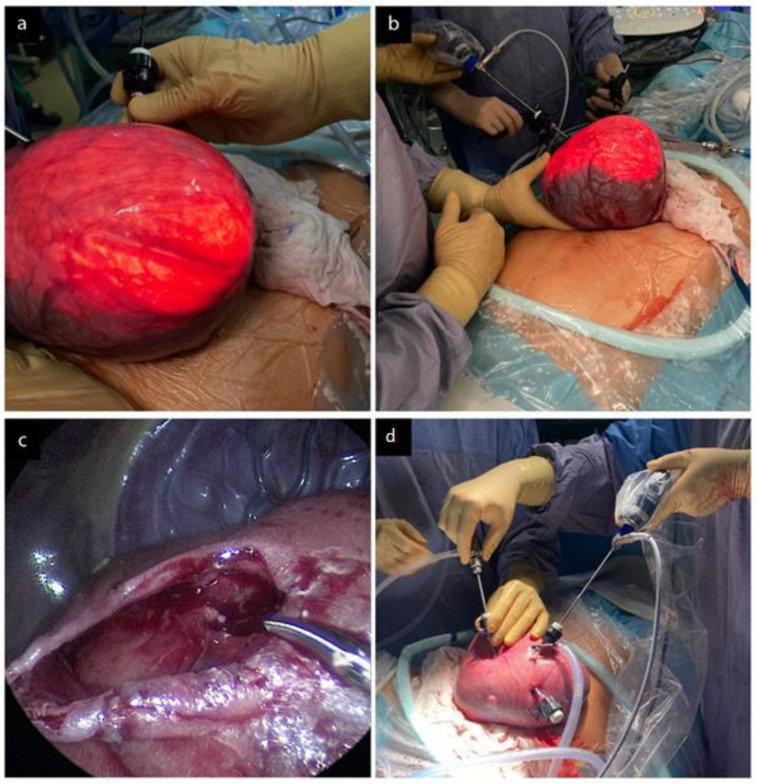
Laparotomy-assisted fetoscopic MMC repair: (**a**) Exteriorized uterus with the first port. The uterine cavity is infused with humidified CO_2_; (**b**) Optical ports are placed in the uterus; fetal positioning; (**c**) Visualization of the opened MMC with the skin-muscle patch. The grasper points to the medullary cone (TSC) positioned at L5 (performed by A. Pastuszka, M.D., Ph.D.); (**d**) Three ports are placed in the uterus during fetoscopic MMC repair.

**Figure 4 diagnostics-12-02978-f004:**
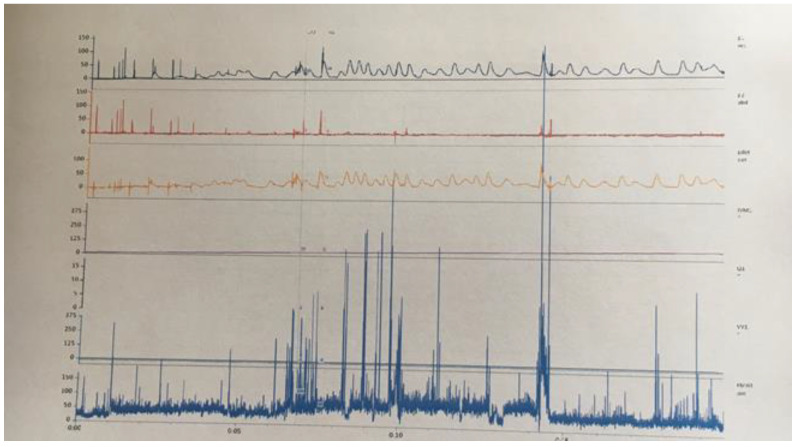
Urodynamic examination: unstable detrusor with hyper-reflexia. High-pressure bladder, bladder capacity correlated with age (80 mL). Leak Point Pressure at 25 mL volume: Pdet/Pves to 48 cm H_2_O, at 38 mL volume: Pdet/Pves to 60 cm H_2_O.

**Figure 5 diagnostics-12-02978-f005:**
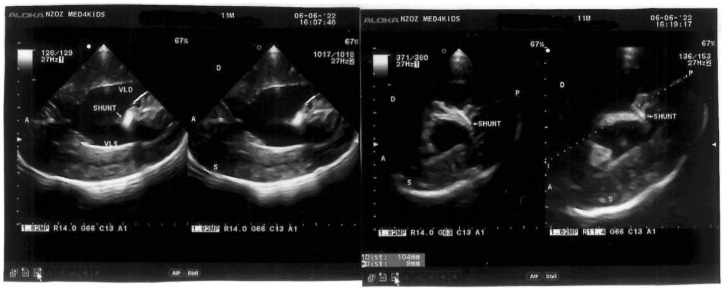
Transcranial ultrasound image of the patient at 11 months postdelivery. (**Right**): Evans index: 0.35. Dilated occipital horn of the lateral ventricles. Ventricular septal defect. Corpus callosum hypoplasia. Subarachnoid cranio-cortical spaces with fluid space. (**Left**): Ventriculoperitoneal shunt passes through the right lateral ventricle below its body and is located in the left lateral ventricle.

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
