# Peer review of "Fetoscopic Myelomeningocele Repair with Complete Release of the Tethered Spinal Cord Using a Three-Port Technique: Twelve-Month Follow-Up—A Case Report"

_diagnostics, 2022, doi:10.3390/diagnostics12122978_

Round 1
Reviewer 1 Report
We should check ethical problems for this paper. Whether has this surgical procedure been approved by IRB or other? This advanced surgical procedure is interesting, and intriguing by pediatric neurosurgeon.
Author Response
We obtained Local Ethics Committee approved of the procedure (no. L.dz.NN-013296/I/02/03).
Information is added to the text in section preoperative procedures.
Reviewer 2 Report
The authors reported a case report of a fetus with a tethered cord & myelomeningocele and that was treated fetoscopically using a three-port technique. I congratulate the authors for the successful management of their case. However, there were multiple issues with the paper that could require substantial improvement.
Title:
* According to the CARE guidelines, it is recommended that the author mention “Case Report” in the title.
Abstract:
* The author mentioned, "24.2WG". Would you please spell the abbreviations when first mentioned in the abstract?
* According to CARE guidelines, it is recommended that the authors mention in the abstract, what is unique about this case. What does it add to the literature?
Introduction:
* The author mentioned, "Up until 1997, surgical correction of the defect within the first 48 hours of neonatal life remained the only repair method for open spina bifida." Would you please add a reference for this sentence?
* The author mentioned, "The outcome left much to be desired as the treatment was associated with progressing intrauterine development of Chiari type II malformation, which in turn led to progressing fetal ventriculomegaly and hindbrain herniation." Would you please add a reference for this sentence?
* The author mentioned, "postnatal OSB repair is associated with the development of shunt-dependent hydrocephalus in 80% of the cases, which requires either ventriculoperitoneal (VP) shunting, to drain excess cerebrospinal fluid (CSF) from the brain ventricles and transport it to the peritoneal cavity, or an endoscopic third ventriculostomy (ETV)." Would you please add a reference for this sentence?
* The author mentioned, "Each of the abovementioned methods is as-sociated with the risk for additional complications such as central nervous system infections or rapidly progressing secondary hydrocephalus connected with the disfunction of the newly created route of CSF evacuation." Would you please add the other complications that can happen during infancy? See this reference: Hasanain AA, Abdullah A, Alsawy MFM, Soliman MAR, Ghaleb AA, Elwy R, Ezzat AAM, Al Menabbawy A, Marei AA, Abd El Razik B, El Hamaky MI, Schroeder HWS, Fleck S, El Damaty A, Marx S, Nowak S, Baldauf J, Zohdi A, El Refaee EA. Incidence of and Causes for Ventriculoperitoneal Shunt Failure in Children Younger Than 2 Years: A Systematic Review. J Neurol Surg A Cent Eur Neurosurg. 2019 Jan;80(1):26-33. doi: 10.1055/s-0038-1669464. Epub 2018 Dec 3. PMID: 30508865.
* The author mentioned, "Towards the end of the previous century, Tulipan and Bruner, of the Vanderbilt University in Nashville, published their reports on fetoscopic repair of open spina bifida." Would you please add a reference for this sentence?
* According to CARE guidelines, it is recommended that the authors mention in the introduction, what is unique about this case. What does it add to the literature?
Case Report:
* Would you please add more details on the patients’ demographics?
* The author mentioned, "Ultrasound evaluation of the fetus revealed the lemon sign, normal division of the forebrain, normal cerebral falx; width of the lateral ventricles: AD left side 19.6, AD right side 15.7; CSP – _4.8 mm, cisterna magna – _3.5 mm, no signs of ACC. Hindbrain ultrasound revealed the banana sign and stage I herniation (Sutton Scale)[12]. Spinal imaging demonstrated a 14mm long myelomeningocele (MMC), upper point of the defect was positioned at L4, with a hernia sack (22x19 mm). Axial view found the lower extremity joints (hip, knee and tarsal) to be normal, with no signs of genu varum or genu valgum; motor function of all three joints was preserved." Would you please add these images?
* The author mentioned, "MRI SSFSET2". Would you please spell the abbreviations when first mentioned in the manuscript?
Discussion
* Would you please a reference to the following sentences?
1- Open spinal dysraphism is a complex defect which develops during the initial stages of the embryonic life and results in the development of Chiari type II malformation.
2- Mid- and hindbrain abnormalities remain the key signs of damage to the central nervous system of the developing fetus.
3- It is caused by insufficient untethering of the neural structures during the repair and secondary tethering of the placode and the spinal nerves in the postoperative scars.
4- Secondary fixed low position of the medullary cone leads to mechanical damage to the over-stretched neural structures during fetal and neonatal life.
* Would you please add the study limitations to the discussion?
General:
* The level of the English language is poor and there are too many errors to identify individually in this revision. Hence, a revision by a professional is highly recommended.
Author Response
Dear Reviewer,
Thank you for all your invaluable comments and suggestions.
Title:
- According to the CARE guidelines, it is recommended that the author mention “Case Report” in the title.
The term ‘case report’ has been added to the title.
=======================================================
Abstract:
- The author mentioned, "24.2WG" (week of gestation or gestational age). Would you please spell the abbreviations when first mentioned in the abstract?
We have indeed neglected to spell the abbreviations in the Abstract – full terms have been added where needed.
=======================================================
- According to CARE guidelines, it is recommended that the authors mention in the abstract, what is unique about this case. What does it add to the literature?
The reference has been added
======================================================================
Introduction:
- The author mentioned, "Up until 1997, surgical correction of the defect within the first 48 hours of neonatal life remained the only repair method for open spina bifida." Would you please add a reference for this sentence?
The reference has been added.
.
==========================================================
- The author mentioned, "The outcome left much to be desired as the treatment was associated with progressing intrauterine development of Chiari type II malformation, which in turn led to progressing fetal ventriculomegaly and hindbrain herniation." Would you please add a reference for this sentence?
The reference has been added.
======================================================
- The author mentioned, "postnatal OSB repair is associated with the development of shunt-dependent hydrocephalus in 80% of the cases, which requires either ventriculoperitoneal (VP) shunting, to drain excess cerebrospinal fluid (CSF) from the brain ventricles and transport it to the peritoneal cavity, or an endoscopic third ventriculostomy (ETV)." Would you please add a reference for this sentence?
The reference has been added. =======================================================
- The author mentioned, "Each of the abovementioned methods is associated with the risk for additional complications such as central nervous system infections or rapidly progressing secondary hydrocephalus connected with the disfunction of the newly created route of CSF evacuation." Would you please add the other complications that can happen during infancy? See this reference: Hasanain AA, Abdullah A, Alsawy MFM, Soliman MAR, Ghaleb AA, Elwy R, Ezzat AAM, Al Menabbawy A, Marei AA, Abd El Razik B, El Hamaky MI, Schroeder HWS, Fleck S, El Damaty A, Marx S, Nowak S, Baldauf J, Zohdi A, El Refaee EA. Incidence of and Causes for Ventriculoperitoneal Shunt Failure in Children Younger Than 2 Years: A Systematic Review. J Neurol Surg A Cent Eur Neurosurg. 2019 Jan;80(1):26-33. doi: 10.1055/s-0038-1669464. Epub 2018 Dec 3. PMID: 30508865.
Other complications which can happen during infancy include intraventricular hemorrhage, proximal occlusion of the shunting system, and inadequate peritoneal CSF absorption capacity.
Information about other complications has been added to the manuscript.
========================================================
- The author mentioned, "Towards the end of the previous century, Tulipan and Bruner, of the Vanderbilt University in Nashville, published their reports on fetoscopic repair of open spina bifida." Would you please add a reference for this sentence?
The reference has been added.
===========================================================
- According to CARE guidelines, it is recommended that the authors mention in the introduction, what is unique about this case. What does it add to the literature?
The explanation has been included in the Introduction and the Abstract section of the manuscript.
To the best of our knowledge, it has been the first neurosurgical fetoscopic untethering of the tethered spinal cord using only the fetal tissues. So far, fetoscopic repairs have been made using the ‘patch’ or ‘patch and skin’ methods.
Case Report:
- Would you please add more details on the patients’ demographics?
maternal age - 29 years, no history of drinking and/or drug abuse, no co-existing morbidities.
====================================================
- The author mentioned, "Ultrasound evaluation of the fetus revealed the lemon sign, normal division of the forebrain, normal cerebral falx; width of the lateral ventricles: AD left side 19.6, AD right side 15.7; CSP – _4.8 mm, cisterna magna – _3.5 mm, no signs of ACC. Hindbrain ultrasound revealed the banana sign and stage I herniation (Sutton Scale)[12]. Spinal imaging demonstrated a 14mm long myelomeningocele (MMC), upper point of the defect was positioned at L4, with a hernia sack (22x19 mm). Axial view found the lower extremity joints (hip, knee and tarsal) to be normal, with no signs of genu varum or genu valgum; motor function of all three joints was preserved." Would you please add these images?
Regretfully, we do not have these images.
=============================================================
- The author mentioned, "MRI SSFSET2". Would you please spell the abbreviations when first mentioned in the manuscript?
The full form - T2-Weighted, Single Shot, Fast Spin Echo Magnetic Resonance Imaging – has been added.
========================================================
Discussion
* Would you please add a reference to the following sentences?
- Open spinal dysraphism is a complex defect which develops during the initial stages of the embryonic life and results in the development of Chiari type II malformation.
[reference no 1].
- Mid- and hindbrain abnormalities remain the key signs of damage to the central nervous system of the developing fetus.
[reference no 10].
- It is caused by insufficient untethering of the neural structures during the repair and secondary tethering of the placode and the spinal nerves in the postoperative scars.
[reference no 17]
4- Secondary fixed low position of the medullary cone leads to mechanical damage to the over-stretched neural structures during fetal and neonatal life.
[reference no 17]
The references have been added.
* Would you please add the study limitations to the discussion?
As it was a case study - not an analysis - we did not include study limitations.
General:
* The level of the English language is poor and there are too many errors to identify individually in this revision. Hence, a revision by a professional is highly recommended.
As per your request, the manuscript has been proofread and corrected by a professional.
Cerification Date: Ufi1 12022 lnvoice number: 07 nal2022
To whom it may concern:
Cerificate of Editing
This letter shall serve as an official ceńification of professional editing and proofreading services. The document detailed below was edited by an experienced editor,
Title: Fetoscopic Myelomeningocele Repair with Gomplete Release of the - Tethered Spinal Cord Using a Three_Poń Technique: Twelve-Month Follow-up
a case repoń.
Author(s}: Agnieszka Pastuszka, Mateusz Zamłyński, Tomasz Horzelski, Jacek Zamłyński, Ewa Horzelska, lwona Maruniak-Chudek, Adriana Marzec, Justyna Paprocka, Patrycja Gazy, Tomasz Koszutski, and Anita Olejek
Editor: lzabella Mrugalska, medical translator and lecturer at the English College, Adam Mickiewicz University
Language: English (US)
This document is certified to have been edited for proper language, style, punctuation, spelling, and grammar, The text as edited is grammatically correct. The authors have accepted the changes to the document.
lf you have any questions, do not hesitate to contact us at Medical Translator services
i. mrugalska@tlumaczmedyczny. com

Round 2
Reviewer 2 Report
Thank you for addressing all my comments in the response letter; however, I could not find these revisions in the manuscript itself.
Would you please change it in the manuscript?
Author Response
We have updated the manuscript, thank you for noticing
Round 3
Reviewer 2 Report
I would like to thank the authors for addressing all my comments.